# CAN DISCRETE INFORMATION EXTRACTION PROMPTS GENERALIZE ACROSS LANGUAGE MODELS?

**Nathanaël Carraz Rakotonirina,**[1] **Roberto Dessì,**[1,2] **Fabio Petroni,**[3] **Sebastian Riedel,**[4] **Marco Baroni**[1,5]
[1]Universitat Pompeu Fabra, [2]Meta AI, [3]Samaya AI, [4]University College London, [5]ICREA

## ABSTRACT

We study whether automatically-induced prompts that effectively extract information from a language model can also be used, out-of-the-box, to probe other language models for the same information. After confirming that discrete prompts induced with the AutoPrompt algorithm outperform manual and semi-manual prompts on the slot-filling task, we demonstrate a drop in performance for AutoPrompt prompts learned on a model and tested on another. We introduce a way to induce prompts by mixing language models at training time that results in prompts that generalize well across models. We conduct an extensive analysis of the induced prompts, finding that the more general prompts include a larger proportion of existing English words and have a less order-dependent and more uniform distribution of information across their component tokens. Our work provides preliminary evidence that it's possible to generate discrete prompts that can be induced once and used with a number of different models, and gives insights on the properties characterizing such prompts.[1]

## 1 INTRODUCTION

NLP has shifted to a paradigm where very large pre-trained language models (LMs) are adapted to downstream tasks through relatively minor updates (Bommasani et al., 2021; Liu et al., 2021). In the most extreme case, task adaptation does not require modifying the LM or even accessing its internals at all, but simply formulating a linguistic query that elicits an appropriate, task-specific response by the model (Petroni et al., 2019a; Radford et al., 2019). This has promising practical applications, as one could easily imagine proprietary LMs only exposing a natural-language-based interface, with downstream agents extracting the information they need by formulating the appropriate queries.[2] In this scenario, one fundamental question is how *robust* the querying protocol is to changes in the underlying LM. On the one hand, the same downstream agent might want to query multiple LMs. On the other, if the LM provider updates the model, this should not break the downstream pipeline. On a more theoretical level, the properties of an emergent robust protocol might give us insights on the general language processing capabilities of neural networks, and how they relate to natural language.

We present a systematic study of the extent to which LM query protocols, that, following current usage, we call *prompting methods*, generalize across LMs. Extending and confirming prior results, we find that discrete prompts that are automatically induced through an existing optimization procedure (Shin et al., 2020) outperform manually and semi-manually crafted prompts, reaching a good performance level *when tested with the same LM used for prompt induction*. While the automatically induced discrete prompts also generalize better to other LMs than (semi-)manual prompts and currently popular "soft" prompts, their overall generalization performance is quite poor. We next show that a simple change to the original training procedure, namely using more than one LM at prompt induction time, leads to discrete prompts that better generalize to new LMs. The proposed procedure, however, is brittle, crucially relying on the "right" choice of LMs to mix at prompt induction. We finally conduct the first extensive analysis of automatically induced discrete prompts,

---

[1]The code to reproduce our analysis is available at `https://github.com/ncarraz/prompt_generalization`.

[2]As a concrete example, one of the most powerful current LMs, GPT3, is only available via a text-based API (`https://beta.openai.com/overview`).

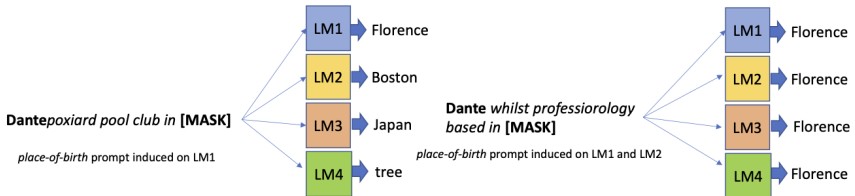

Figure 1: Cartoon summary of our main results. Prompts induced using a single language model have a significant drop of performance when used to query other models. The problem is alleviated when prompts are exposed to multiple models in the induction phase. Subtle but consistent differences in the nature of the induced prompts also emerge.

tentatively identifying a set of properties characterizing the more general prompts, such as a higher incidence of existing English words and robustness to token shuffling and deletion.

## 2 RELATED WORK

Prior work such as Petroni et al. (2019a) and Radford et al. (2019) demonstrated that LMs can be directly adapted to new tasks through appropriate querying methods. This led to an explosion of work on so-called "prompt engineering" (see Liu et al., 2021, for a thorough review). Much of this work focuses on crafting appropriate manual or semi-manual prompts and/or on tuning LMs to better respond to such prompts (e.g., Schick & Schütze, 2021; Sanh et al., 2022). Going beyond manual prompts, Shin et al. (2020) introduced the AutoPrompt algorithm to generate prompts using gradient-guided search, and demonstrated that such prompts often outperform manual ones. While automatically induced prompts suffer of issues such as low-interpretability, we think it is important to continue focusing on them because, besides their better performance (a result we confirm here for AutoPrompt across a range of LMs), they are more promising than manual prompts in terms of scalability, especially in contexts in which it is not sufficient to formulate a single prompt template for a whole task, but each input query demands a distinct prompt formulation (Zhang et al., 2022).

Concurrent and later work has proposed to replace discrete strings, such as those generated by AutoPrompt, with sequences of arbitrary vectors from the LM's embedding space (Lester et al., 2021; Zhong et al., 2021). We confirm here that these continuous, or "soft" prompts outperform AutoPrompt when trained and tested on the same LM. However, they cannot be used in our envisaged multiple-LM scenario. First, they require access to a model inner representations, beyond the standard natural language querying interface, so that embeddings can be passed as input. Second, continuous prompts, by their nature, won't generalize out-of-the-box to other LMs. Trivially, they can't generalize across models with different embedding dimensionality. Even when models share dimensionality, there is no reason why the absolute position of a vector in the embedding space of a model should meaningfully transfer to another model. Discretizing soft prompt tokens to their nearest vocabulary neighbours in order to overcome these issues does not help either. Khashabi et al. (2021) demonstrated that it is possible to find well-performing soft prompts whose nearest neighbor projections are arbitrarily fixed discrete tokens. Appendix B elaborates on the failure of soft prompts to generalize across models, as well as the problematic behaviour of discretized soft prompts.

We are not aware of much previous work that has addressed the challenge of LM-to-LM transferability. Wallace et al. (2019) studied this problem in the context of textual adversarial attacks (that can be seen as a special case of prompting, and indeed their attack method is closely related to AutoPrompt). Similarly to us, they notice some performance drop when transferring adversarial "triggers" to different LMs, and they show that this can be mitigated by an ensembling approach where two triggers generated using variants of the same LM are combined.

Su et al. (2022) study LM-to-LM transferability in the context of continuous prompts. Since, as we just discussed, such prompts are not directly transferable, they induce a projection from the embedding space of the source LM to that of the target LM, thus considering a very different scenario from the type of "out-of-the-box" transferability we are interested in here.

# 3 EXPERIMENTAL SETUP

## 3.1 DATA

We focus on the task of slot-filling which, since its introduction in LM evaluation through the LAMA benchmark (Petroni et al., 2019a), has been extensively used to probe the knowledge contained in LMs (AlKhamissi et al., 2022). More specifically, we use the T-ReX split (Elsahar et al., 2018) of LAMA. Each fact in T-ReX is represented as a triple $\langle subject, relation, object \rangle$—for example, $\langle Dante, place\ of\ birth, Florence \rangle$. LMs are queried using cloze prompts as in "*Dante* was born in ______." A LM is said to have properly stored a fact if it can successfully predict the ground-truth object given a prompt and the corresponding subject. We have decided to focus primarily on this task because the prompts convey actual semantic information (characterizing the relation between the subject and the object) rather than just metalinguistic information (as would be the case, for example, for a machine-translation prompt, which might express something like: "translate the next sentence from English to French"). Furthermore, the task requires learning a different prompt for each relation, which can be seen as a first step toward fully flexible prompts that would change with each single input (Zhang et al., 2022).

The LAMA test set contains 41 relations, each with up to 1,000 facts. We also evaluate on the more challenging LAMA-UHN subset (Poerner et al., 2019), addressing some of the weaknesses of the original LAMA, in Appendix D. All prompting methods are trained using the training data collected by Shin et al. (2020), which include 1,000 facts for each relation type, drawn from either the original T-REx dataset or Wikidata. LAMA also provides manual prompts, which we will use in our experiments.

Since each LM class in our experiments (see Section 3.2 below) has its own vocabulary, a common subset must be used for fair comparison. This is obtained from the intersection of the vocabularies of all models considered. Furthermore, the training and test datasets are filtered to ensure that each object is included in the common vocabulary. There are 11,511 case-sensitive items in the common vocabulary. The filtered test set contains 25,358 facts, while the training set contains 25,247 facts.

We evaluate prompting methods using micro-averaged accuracy (precision@1).

## 3.2 LANGUAGE MODELS

Our experiments cover the three main types of LM. We use pre-trained LMs without any kind of parameter updating. Table 1 shows the LMs considered in this study.

**Masked LMs**   They produce representations using both the left and right context. Given a sequence $\boldsymbol{x} = [x_1, ..., x_n]$, they estimate the probability of a token $\boldsymbol{x}_i$ given its left and right context $p(\boldsymbol{x}_i) = p(x_i|x_1, ..., x_{i-1}, x_{i+1}, ..., x_n)$.

**Left-to-right LMs**   They predict the next token conditioned on previous ones or assign a probability to a sequence of tokens. Given a sequence of tokens $\boldsymbol{x} = [x_1, ..., x_n]$, left-to-right LMs assign a probability $p(\boldsymbol{x})$ to the sequence using the chain rule $p(\boldsymbol{x}) = \prod_t p(x_t|x_1, ..., x_{t-1})$.

**Sequence-to-sequence LMs**   They are composed of a bidirectional encoder that uses both left and right context and a left-to-right decoder that do not share parameters.

## 3.3 PROMPT INDUCTION METHODS

Prompts are either manually crafted or generated automatically by prompting methods. In this study, we have selected 3 different prompting methods that are representative of semi-manual, discrete and continuous induction methods, respectively. They have been shown to perform well on the slot filling task and associated code is publicly available.

**LPAQA**   Starting from seed manual prompts, Jiang et al. (2020) generate a diverse candidate prompt set using mining- and paraphrasing-based methods. For each relation, the best performing candidate on the training data is selected. To improve performance, the authors also propose

Table 1: Pre-trained language models considered in this study.

| Model | Type | #Parameters | Training Corpus |
|---|---|---|---|
| BERT$_{BASE}$ (Devlin et al., 2019) | Masked | 110M | |
| BERT$_{LARGE}$ (Devlin et al., 2019) | Masked | 340M | |
| DistilBERT (Sanh et al., 2019) | Masked | 66M | Wikipedia (en) & BookCorpus (16GB) |
| RoBERTa$_{BASE}$ (Liu et al., 2019) | Masked | 125M | Wikipedia (en) & BookCorpus & CC-News |
| RoBERTa$_{LARGE}$ (Liu et al., 2019) | Masked | 355M | & OpenWebText & Stories (160GB) |
| DistilRoBERTa (Sanh et al., 2019) | Masked | 82M | OpenWebText (38GB) |
| GPT2 (Radford et al., 2019) | Left-to-right | 117M | |
| GPT2$_{MEDIUM}$ (Radford et al., 2019) | Left-to-right | 345M | |
| GPT2$_{LARGE}$ (Radford et al., 2019) | Left-to-right | 774M | WebText (40GB) |
| GPT2$_{XL}$ (Radford et al., 2019) | Left-to-right | 1.5B | |
| BART$_{BASE}$ (Lewis et al., 2019) | Seq2seq | 140M | Wikipedia (en) & BookCorpus & CC-News |
| BART$_{LARGE}$ (Lewis et al., 2019) | Seq2seq | 400M | & OpenWebText & Stories (160GB) |
| T5$_{SMALL}$ (Raffel et al., 2020) | Seq2seq | 60M | |
| T5$_{BASE}$ (Raffel et al., 2020) | Seq2seq | 220M | C4 & Wiki-DPR (765GB) |
| T5$_{LARGE}$ (Raffel et al., 2020) | Seq2seq | 770M | |

prompt ensembling. However, ensembling tends to increase performance independently of the underlying prompting method (see Appendix A). Consequently, we will only focus on the top-1 prompt selection method here. We consider LPAQA a semi-manual method because it needs to be seeded with manual prompts, and mining retrieves further human-generated strings.

**AutoPrompt** It is an automated method proposed by Shin et al. (2020) to generate discrete prompts using gradient-guided search (Wallace et al., 2019). The prompts are composed of a sequence of tokens selected from the vocabulary of the LM. The number of tokens is pre-defined. The process is divided into two phases. For each specific token position, a set of candidates that maximize the likelihood on a batch of training data is first created. Then, the candidates are re-evaluated on a different batch, with the best one retained. Even though the generated prompts are less interpretable, they perform better than manual prompts. In our experiments, we use 5-token prompts and run the algorithm for 1,000 iterations.

**OptiPrompt** Zhong et al. (2021) propose an automated method to generate continuous prompts. They are dense vectors in the embedding space of the LM that are learned using gradient descent on a separate training dataset. Except for the learning rate, which is increased to 3e-2 for the T5 models for proper convergence, we use the same hyperparameters as the original implementation. We initialize vectors randomly.

## 4    RESULTS AND ANALYSIS

### 4.1    PROMPTING METHOD PERFORMANCE

We start by evaluating the performance of the different prompting methods.[3] Prompts are induced with a specific LM and then evaluated by retrieving objects from the same LM. Table 2 summarizes the results. For reference, a majority-class baseline always picking the most common object for each relation reaches 26.91% accuracy (note that this baseline has partial access to the ground truth in order to retrieve the most common object of each relation). The random baseline is virtually at 0%.

AutoPrompt clearly outperforms LAMA and LPAQA, although it lags behind OptiPrompt. We thus confirm that soft prompts are the way to go if you have access to a model embedding space and are not interested in generalization across models. If either of these conditions is not met, AutoPrompt is preferable to manual and semi-manual prompts.

Masked models tend to perform better as source LMs. This could be attributed to the fact that they were pre-trained with a fill-in-the-blank task (Devlin et al., 2019), which is exactly how the slot filling task is formulated.

---

[3]Following Petroni et al. (2019b), when testing manual and LPAQA prompts with left-to-right LMs, only the tokens before [MASK] are used. [MASK] is always the last AutoPrompt and OptiPrompt token.

Table 2: Comparison of different prompting methods using micro-averaged accuracy (precision@1).

| Model | LAMA | LPAQA | AutoPrompt | OptiPrompt |
|---|---|---|---|---|
| BERT$_{\text{BASE}}$ | 34.82 | 41.18 | **50.09** | 48.26 |
| BERT$_{\text{LARGE}}$ | 35.81 | 42.15 | 49.52 | **50.88** |
| DistilBERT | 6.75 | 13.14 | 29.79 | **44.76** |
| RoBERTa$_{\text{BASE}}$ | 26.36 | 32.80 | 39.63 | **44.73** |
| RoBERTa$_{\text{LARGE}}$ | 31.63 | 40.54 | 44.12 | **47.39** |
| DistilRoBERTa | 23.80 | 32.43 | 41.17 | **44.21** |
| GPT2 | 7.23 | 9.63 | 11.36 | **39.28** |
| GPT2$_{\text{MEDIUM}}$ | 13.74 | 18.29 | 18.59 | **38.43** |
| GPT2$_{\text{LARGE}}$ | 15.50 | 19.97 | 12.91 | **44.14** |
| GPT2$_{\text{XL}}$ | 16.98 | 21.31 | 15.42 | **47.76** |
| BART$_{\text{BASE}}$ | 22.95 | 32.43 | 39.63 | **43.05** |
| BART$_{\text{LARGE}}$ | 27.07 | 36.78 | 26.56 | **45.06** |
| T5$_{\text{SMALL}}$ | 14.94 | 20.94 | **31.44** | 28.10 |
| T5$_{\text{BASE}}$ | 24.35 | 32.70 | 39.59 | **40.51** |
| T5$_{\text{LARGE}}$ | 28.21 | 36.07 | 42.00 | **44.42** |
| average | 22.00 | 28.69 | 32.62 | **43.39** |
| st dev | 9.17 | 10.56 | 13.12 | 5.40 |

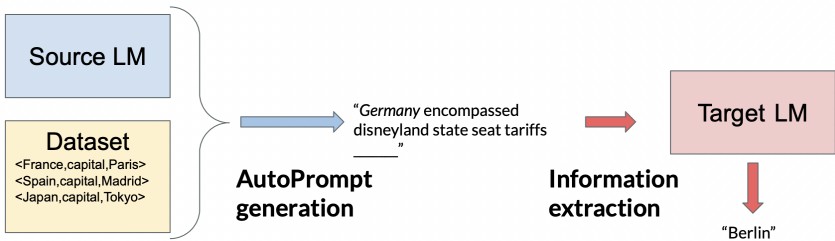

Figure 2: Prompt generalization. For each relation, a prompt is induced from the *source* LM. At test time, the prompt is tested on the *target* LM.

## 4.2 AUTOPROMPT GENERALIZATION

In this section, we investigate AutoPrompt's ability to generalize across different LMs.[4] Prompts are induced using a *Source* LM and then evaluated on a *Target* LM, which can be the same or different from the Source (Figure 2).

The results, relative to single-model LM performance, are shown in Figure 3. AutoPrompt generally performs best when the Source and Target LMs are the same, as shown by the fact that off-diagonal values are mostly negative. The performance gap gets bigger as we transfer across different LM types. Prompts are generally more stable across different sizes of the same model, such as BERT$_{\text{BASE}}$ and BERT$_{\text{LARGE}}$. The drop in generalization performance of left-to-right models is less dramatic simply because, for these models, the original same-source-and-target performance is already low.

We also verify the impact of model size on generalization. For each source model, we define the generalization drop score as the average of the corresponding column in Figure 3. It measures the average drop in accuracy when prompts from a source model are tested on a different target, with respect to the original same-source-and-target accuracy. We discovered that performance drop and source model size are highly correlated (0.6). We do not have a clear explanation for this correlation, but we think it is an intriguing observation that should be further studied in the prompt analysis literature.

---

[4]Appendix B confirms that OptiPrompt prompts do not generalize well. Appendix C shows that AutoPrompt outperforms LPAQA also in terms of generalization.

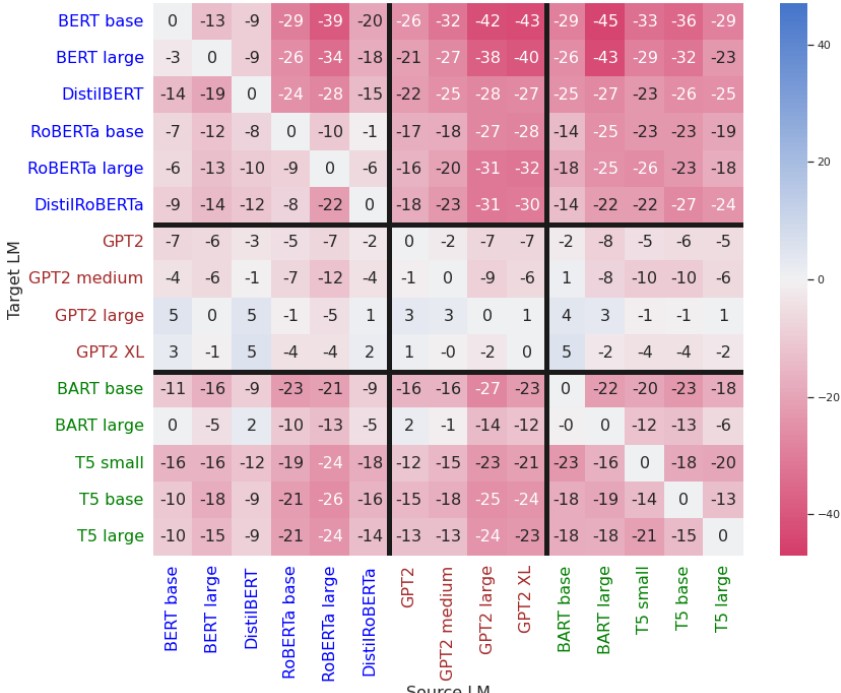

Figure 3: AutoPrompt relative performance across LMs. Each column represents a Source LM and each row a Target LM. Each value represents the difference between the accuracy achieved by the Target LM when using prompts induced with the Source LM and the accuracy obtained when Target is also used for training.

## 4.3 MIXED-TRAINING AUTOPROMPT GENERALIZATION

We propose a simple modification to AutoPrompt training to generate prompts that generalize better. Recall that the AutoPrompt algorithm involves two phases: one in which candidate prompts are generated, and one in which the prompts are evaluated. Rather than relying on the same model for the two phases, we now use two different LMs. The first model, which we call the *generator*, proposes a set of candidates. Then, the second model, that we call the *evaluator*, evaluates the candidates and chooses the best one.

To avoid a combinatorial explosion, we focus on combining the single LMs of each class that performed best in the same-source-and-target setup (Table 2 above). For each pair, we arbitrarily use the best of the two LMs (in the same-source-and-target setup) as generator. We deliberately avoid picking models based on generalization performance (Figure 3), as in a realistic settings we might not have advance access to the new architectures we need to generalize to.

Table 3 compares the performance of standard and mixed AutoPrompt (we extend these experiments to LAMA-UHN in Appendix D). The BERT$_{BASE}$/T5$_{LARGE}$ mix has the highest average accuracy. Although it does not perform as well as BERT$_{BASE}$ on the BERT models, it transfers better to all the other models (including the RoBERTa family). This mixed AutoPrompt variant even outperforms the best seq2seq model T5$_{LARGE}$ on all sequence-to-sequence models (including T5$_{LARGE}$ itself). It also outperforms GPT2$_{MEDIUM}$ on two GPT2 variants.

If these results are very encouraging, Table 3 also shows that simply mixing models does not guarantee good generalization. When we replace BERT$_{BASE}$ with T5$_{LARGE}$ as generator, generalization performance is actually *worse* than when using T5$_{LARGE}$ alone, and combining BERT$_{BASE}$ as generator with GPT2$_{MEDIUM}$ as evaluator leads to minor generalization improvements compared to using BERT$_{BASE}$ alone. Some preliminary insights on the best mixing strategy are offered in Appendix E.

Table 3: AutoPrompt mixed training. The first three columns report generalization accuracy for the single best LM in each class; the next three columns evaluate their combination.

| Source / Target | BERT$_{BASE}$ | GPT2$_{MEDIUM}$ | T5$_{LARGE}$ | BERT$_{BASE}$/T5$_{LARGE}$ | BERT$_{BASE}$/GPT2$_{MEDIUM}$ | T5$_{LARGE}$/GPT2$_{MEDIUM}$ |
|---|---|---|---|---|---|---|
| BERT$_{BASE}$ | **50.09** | 18.02 | 20.83 | 41.13 | 38.64 | 16.47 |
| BERT$_{LARGE}$ | **47.01** | 22.54 | 26.43 | 40.51 | 39.6 | 16.29 |
| DistilBERT | **15.75** | 4.37 | 4.71 | 15.08 | 13.35 | 3.65 |
| RoBERTa$_{BASE}$ | 32.31 | 21.31 | 20.28 | **36.01** | 30.56 | 17.24 |
| RoBERTa$_{LARGE}$ | 37.79 | 24.07 | 26.06 | **38.63** | 34.24 | 21.16 |
| DistilRoBERTa | 31.71 | 18.28 | 17.24 | **33.49** | 28.53 | 16.58 |
| GPT2 | 4.70 | 9.71 | 6.04 | **10.19** | 7.53 | 8.12 |
| GPT2$_{MEDIUM}$ | 14.38 | 18.59 | 12.51 | 16.29 | **22.49** | 16.47 |
| GPT2$_{LARGE}$ | 17.95 | 15.68 | 13.52 | **22.33** | 19.95 | 14.84 |
| GPT2$_{XL}$ | 18.34 | 15.02 | 13.09 | 19.74 | **23.19** | 18.87 |
| BART$_{BASE}$ | 28.98 | 24.11 | 21.62 | **34.57** | 31.62 | 18.84 |
| BART$_{LARGE}$ | 26.73 | 25.20 | 20.32 | **33.73** | 29.15 | 18.94 |
| T5$_{SMALL}$ | 15.78 | 16.23 | 11.89 | **19.31** | 18.28 | 7.99 |
| T5$_{BASE}$ | 29.26 | 22.04 | 26.58 | **36.67** | 32.06 | 16.99 |
| T5$_{LARGE}$ | 32.32 | 28.90 | 42.00 | **44.51** | 35.71 | 27.22 |
| Average | 26.87 | 18.94 | 18.88 | **29.48** | 26.99 | 15.98 |

| training LM(s) | semantic overlap | real-word ratio | shuffled accuracy non-normalized | ratio |
|---|---|---|---|---|
| **BERT$_{BASE}$** | 5.3* | 81.7 | 11.5 (3.1) | 23.0 (6.2) |
| **GPT2$_{MEDIUM}$** | 0.97 | 68.8 | 6.0 (1.2) | 32.4 (6.7) |
| **T5$_{LARGE}$** | 2.43* | 71.9 | 15.1 (2.6) | 36.0 (6.1) |
| **BERT$_{BASE}$/T5$_{LARGE}$** | 3.29* | 86.0 | 11.5 (3.1) | 27.9 (7.4) |
| **BERT$_{BASE}$/GPT2$_{MEDIUM}$** | 3.51* | 88.6 | 9.5 (3.1) | 24.5 (8.0) |
| **T5$_{LARGE}$/GPT2$_{MEDIUM}$** | 1.44 | 73.3 | 9.7 (2.4) | 35.8 (9.0) |

Table 4: AutoPrompt prompt analysis. The *semantic overlap* column reports the $t$-score for the difference in semantic overlap between matching and mismatched prompts (see text for explanation), with * marking significant scores at $\alpha$=0.05. The *real-word ratio* column reports percentage ratios of corpus-attested English words among space-/punctuation-mark delimited tokens appearing in a prompt set. The *shuffled accuracy* columns report percentage accuracy after token shuffling, divided by the original accuracy in the *ratio* column (averages of 10 random shufflings with standard deviations in parenthesis).

## 4.4 PROMPT ANALYSIS

We study the prompts generated by AutoPrompt through single-model and mixed training, looking for differences that might hint at how the latter generalize better. Since the prompt examples in Table 8 (Appendix F) suggest that there are no huge differences directly visible to the "naked eye", we undertake a quantitative analysis led by the following hypotheses.

1) Each LM has its own peculiarities, but they all share English as training language. Prompts optimized on a single model might overfit the quirks of that model, but mixed-training prompts might capture more general properties of English that are shared across models. We thus hypothesize that *prompts that generalize better will have a larger semantic overlap with manually crafted English prompts*. 2) Modern LMs use sub-word tokenization strategies that differ from model to model. AutoPrompt is thus free to combine sub-words into non-word sequences (e.g., *slalomgraphers*, *publishedtoon* in Table 8) that might in turn be tokenized differently by different models, leading to inter-model brittleness. We thus conjecture that *prompts that generalize better will contain a larger proportion of real English words*. 3) Natural languages rely on word order to express meaning, but it's less clear that LMs are capturing genuine syntactic rules (Sinha et al., 2021). It's more likely that, to the extent that prompts crucially rely on token order, this is exploiting statistical co-occurrence quirks of specific LMs. We thus conjecture that a "bag-of-token" prompt sequence that does not require the tokens to be in any special order will be more general than one where order matters and, consequently, *generalizing prompts will be more robust to token shuffling*. 4) On a related point, single-model-optimized prompts might concentrate information in the slots the source model is most sensitive to, but such slots might vary from model (type) to model (type). We thus conjecture that generalizing prompts will distribute information more evenly across tokens and thus *they will be more robust to single-token deletion*.

**Semantic overlap with English**    The manual LAMA prompts are our English reference point. We measure semantic overlap as the cosine between a vector representing an AutoPrompt-generated prompt and a LAMA prompt. Specifically, we use *fastText* (Bojanowski et al., 2017) to represent prompts, both because it offers independent representations from those of any of the LMs we are comparing, and because it relies on vocabulary- and tokenization-independent n-gram-based sequence representations. Instead of reporting difficult-to-interpret absolute cosines, we report the $t$-score for the cosine difference between cases where AutoPrompt prompts are compared to the LAMA prompts for the same T-ReX relation, and cases where AutoPrompt prompts are compared to different-relation LAMA prompts. In other words, the larger this value is, the clearer the difference in semantic overlap is between meaningful and random prompt comparisons. Results are in the first column of Table 4. There is a strong correlation ($>0.9$ Pearson) between a prompt semantic overlap with English and its accuracy when tested on the model used as source/generator during training. This is encouraging in itself, suggesting that more effective AutoPrompt prompts are also more semantically transparent. However, our hypothesis that better generalization implies higher semantic overlap is disproven: there is a clear decrease in overlap between $\text{BERT}_{\text{BASE}}$-based prompts and the better generalizing ones obtained through $\text{BERT}_{\text{BASE}}/\text{T5}_{\text{LARGE}}$ mixed-training.

**Real-word ratio**    We verify whether a space- or punctuation-marked delimited character sequence in a prompt is an existing English word by checking if it appears in the list of 56k words occurring at least 1k times in the ukWaC corpus (Baroni et al., 2009). This corpus was not used for training any of the models, thus minimizing biases towards any of them. The minimum occurrence threshold was determined by manual inspection, observing that rarer strings tend not to be real words but "corpus detritus" (numbers, dates, code fragments, typos). The second column of Table 4 reports percentage attested-word ratios among the tokens produced by AutoPrompt for the whole T-ReX relation set in various training setups. For reference, this ratio is at 99.4% for the manual LAMA prompts. The generalizing $\text{BERT}_{\text{BASE}}/\text{T5}_{\text{LARGE}}$ setup clearly outperforms single-model training on $\text{BERT}_{\text{BASE}}$ on this metric, tentatively confirming our hypothesis that more word-like strings will transfer better across models. Note however that $\text{BERT}_{\text{BASE}}/\text{GPT2}_{\text{MEDIUM}}$, a mixed-training setup that does not generalize better than $\text{BERT}_{\text{BASE}}$-based training alone, features prompts sporting an even higher proportion of existing words. So, the latter might be a common property of mixed-training-induced prompts, but not one that automatically entails better generalization.

**Shuffling**    We shuffle the tokens in each prompt, and compute the resulting T-ReX accuracy when retrieving information from the LM used as source/generator during AutoPrompt training. To tease the effect of shuffling apart from the absolute performance of a prompt set, we also report the ratio of accuracy after shuffling to accuracy with unshuffled prompts. We repeat the shuffling experiment 10 times, and report averaged accuracies/ratios and standard deviations in the last two columns of Table 4. By superficially eyeing AutoPrompt prompts such as those in Table 8, one could think they are bags of tokens, but the ratios show that token order matters, as there is always a big drop in performance after shuffling. Indeed, by closer inspection of the prompts in Table 8, we notice that some of them do have a sentence flavor ('*[X] teaches modelling frescoes downtown in [Y]*'). Our hypothesis that the generalizing $\text{BERT}_{\text{BASE}}/\text{T5}_{\text{LARGE}}$ prompts are less order-sensitive than the $\text{BERT}_{\text{BASE}}$ ones is confirmed (compare the ratios of these setups). The extra robustness of $\text{BERT}_{\text{BASE}}/\text{T5}_{\text{LARGE}}$ might be due to the fact that $\text{T5}_{\text{LARGE}}$ itself is particularly robust to shuffling, and it remains to be seen if for mixed-training prompts there is a causal relationship between robustness to shuffling and generalization. Note that the relatively high ratio of $\text{GPT2}_{\text{MEDIUM}}$ might be due to a flooring effect, as this setup has low accuracy before and after shuffling.

**Token deletion**    To verify if information is equally distributed across the 5 tokens of an Auto-Prompt prompt, for each prompt set we compute T-ReX accuracy (retrieving information from the matching source/generator model) after removing a single token, repeating the experiment for each token position. Figure 4 reveals that $\text{BERT}_{\text{BASE}}$-based prompts are concentrating a lot of information towards the end of the sequence, and in particular on the last token, with a huge drop in performance if the latter is deleted. While the same tendency to pack information towards the end of the sequence is present in the better-generalizing $\text{BERT}_{\text{BASE}}/\text{T5}_{\text{LARGE}}$ prompts, the drop is less dramatic (especially when taking into account the fact that the latter prompts start from a lower full-sequence accuracy). Figure 5 in Appendix G shows that all generated prompts have a tendency to pack more information at the end of the sequence (which is remarkable, as their encoders are based on the fully

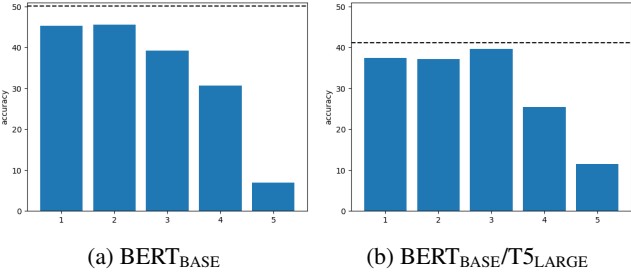

(a) BERTBASE                    (b) BERTBASE/T5LARGE

Figure 4: Percentage accuracy after dropping the token in each position of an AutoPrompt-generated 5-token prompt. The dashed horizontal line marks full-sequence accuracy.

symmetric transformer architecture), but T5$_{\text{LARGE}}$ generates prompts that are less affected by single-token deletion. The higher robustness of BERT$_{\text{BASE}}$/T5$_{\text{LARGE}}$ compared to BERT$_{\text{BASE}}$ might thus be inherited from T5$_{\text{LARGE}}$, and further studies should ascertain whether there is a causal relation between smoother information distribution and better generalization for mixed-training prompts.

## 5 Discussion

**Take-home points**  Automatically induced discrete prompts, such as those derived with the Auto-Prompt algorithm, strike a good balance between (semi-)manual prompts, that they outperform in retrieval quality and (potentially) scalability, and soft prompts, that can only be used out-of-the-box on the model they were induced from, and require access to inner model structures to function. However, the standard AutoPrompt method must be adapted to get good performance across language models. In particular, a simple modification in which AutoPrompt is trained using *two* language models leads to prompts that better generalize to further models.

The better-generalizing prompts induced in this way look quite similar to prompts induced with the standard method. However, a probing analysis suggests that there are systematic differences, and in particular that the generalizing prompts tend to feature a larger proportion of existing English words, and to be more robust to ablations that probe sensitivity to token order and asymmetries in information distribution.

In sum, our results suggest that it is viable to use a learning-based algorithm to generate "universal" discrete prompts that can be employed to extract information from a variety of pre-trained language models, only requiring access to their standard discrete-string interface.

**Limitations and directions for further work**  Our results are based on a single task, slot filling, and a single data-set (T-ReX). We believe this is a good starting point, because in slot filling a separate, semantically contentful prompt must be learned for each relation, but future work should extend the investigation to different tasks, including tasks where successful knowledge retrieval requires more than recalling a single word, as in the LAMA/T-ReX setup we used.

We used a single discrete prompt induction algorithm, AutoPrompt, confirming that it is generating high-quality prompts. However, this method generates a single fixed prompt for each task or sub-task. True scalability will only be achieved with prompting methods that can generate an appropriate query for each different input they receive, and we intend to design discrete-prompt-induction algorithms matching this desideratum (see Haviv et al. (2021) for a step in this direction). Another reason to focus on algorithm development is that the single-model comparison between AutoPrompt and the OptiPrompt soft prompts shows there is still large room to improve retrieval quality.

Our analysis revealed systematic differences between the best model-specific and generalizing prompts. However, it is not clear that any of these differences is causally connected with generalization improvement. In future work, we would like to better understand the relation between properties such as robustness to shuffling and generalization. A particular exciting direction is to favour the emergence of such properties through appropriate auxiliary functions at prompt-induction time, and verify whether they lead to further improvements in generalization performance.

## ACKNOWLEDGMENTS

We thank the reviewers for constructive feedback. We thank the members of the UPF COLT group for discussion and advice. UPF has received funding from the European Research Council (ERC) under the European Union's Horizon 2020 research and innovation programme (grant agreement No. 101019291). This paper reflects the authors' view only, and the funding agency is not responsible for any use that may be made of the information it contains. Fabio Petroni conducted work on the project while at Meta AI.

## REPRODUCIBILITY STATEMENT

All datasets, pre-trained models and prompting methods used in this paper are publicly available. We use the same hyperparameters as the original implementation of the prompting methods unless it is clearly specified in the text. Code to reproduce the results as well as the common vocabulary and the filtered datasets will be shared upon acceptance.

## ETHICS STATEMENT

Prompting relies on pre-trained language models, and thus inherits most ethical risks inherent in such models (Weidinger et al., 2022). As we are not introducing new models, we are not adding new potential issues tied to LMs.

As the examples in Table 8 show, automated prompting methods tend to produce opaque prompts. This characteristic might be exploited for harmful purposes, such as adversarial attacks in which a model is "triggered" to produce unwanted information through an apparently innocuous prompt (Wallace et al., 2019). We believe that this provides further motivation for our focus on discrete prompts, that are more directly human-readable than soft prompts. We showed in particular in Section 4.4 that there is a very high correlation ($>.9$) between the quality of automatically-induced prompts in the relevant information retrieval task and the degree of semantic transparency of the prompts. If this result could be extended, it would constitute very good news on the interpretability front, suggesting that very high-quality prompts will also be more human-interpretable, making it harder to exploit opaque prompts for harmful purposes.

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

## A    PROMPT ENSEMBLING

Discrete prompt ensembling for knowledge extraction was introduced by Jiang et al. (2020) using LPAQA. Instead of selecting the top-1 prompt from a set of candidates, multiple prompts are combined in order to improve performance. The authors argue that ensembling improves the extraction

Table 5: Direct OptiPrompt transfer across LMs with same embedding dimension.

| Source
Target | BERT$_{BASE}$ | DistilBERT | RoBERTa$_{BASE}$ | DistilRoBERTa | GPT2 | BART$_{BASE}$ | T5$_{BASE}$ |
|---|---|---|---|---|---|---|---|
| BERT$_{BASE}$ | **48.25** | 33.71 | 0.62 | 1.24 | 1.02 | 1.00 | 0.19 |
| DistilBERT | 11.89 | **46.41** | 0.97 | 0.83 | 1.64 | 0.81 | 0.64 |
| RoBERTa$_{BASE}$ | 2.27 | 2.02 | **46.17** | 41.04 | 2.67 | 3.13 | 4.90 |
| DistilRoBERTa | 2.12 | 1.88 | 39.86 | **45.26** | 2.48 | 2.87 | 3.81 |
| GPT2 | 0.01 | 0.04 | 0.15 | 0.01 | **40.57** | 0.00 | 0.00 |
| BART$_{BASE}$ | 1.20 | 0.72 | 1.02 | 0.05 | 0.60 | **44.3** | 0.27 |
| T5$_{BASE}$ | 4.17 | 4.53 | 4.89 | 4.64 | 2.89 | 4.92 | **42.08** |

of knowledge that appeared in different contexts within the training data. Since the idea of ensembling is distinct from the specifics of the LPAQA method, in this appendix we check if ensembling also helps AutoPrompt. Specifically we experiment with the optimized ensembling approach used with LPAQA by Jiang et al. (2020), which associates learnable weights to each prompt.

For ensembling purposes, multiple candidate prompts for each relation are required. One way to obtain these candidate prompts for AutoPrompt is by inducing prompts from instances of the same model trained with different seeds. Considering the computational cost of retraining the models ourselves, we use the already available MultiBERT collection (Sellam et al., 2021), which contains 25 instances of BERT$_{BASE}$ trained with different seeds. Unfortunately, unlike the models in Table 1, the MultiBERTs were trained on uncased data, which means we can only evaluate on the MultiBERTs themselves. Specifically, we use seed-0 MultiBERT as our target model, and obtain the AutoPrompt prompts from the remaining MultiBERTs. For LPAQA, we use seed-0 MultiBERT for training and testing, obtaining the candidate prompts with the standard LPAQA mining and paraphrasing procedure.

First, we confirm that in this different setup AutoPrompt is still outperforming LPAQA in top-1 evaluation (39.40% vs. 35.17% T-ReX accuracies, respectively). Second, ensembling helps both models. However, ensembled LPAQA barely reaches the performance of top-1 Automprompt (39.21%), whereas ensembled AutoPrompt further improves to 42.43% accuracy.

## B  SOFT PROMPT GENERALIZATION

Transferring soft prompts across models is not as straightforward as with discrete prompts. In order to transfer the vectors, the source and target must have the same embedding dimension. We experiment with the subset of models that have a common embedding dimension of 768. Table 5 confirms that soft prompts generalize extremely poorly. The only exception is decent student/teacher transfers for distilled models, probably because their embeddings were initialized with those of the corresponding teacher models.

A more general approach consists in discretizing the soft prompts to their nearest vocabulary neighbors (Mikolov et al., 2013; Hashimoto et al., 2016), and using the resulting vocabulary item sequences as prompts. OptiPrompt vectors can be initialized with random numbers, random vocabulary embeddings or manual prompts. When initialized with random vocabulary embeddings or manual prompts, the nearest neighbor projections of the resulting vectors are always identical to the initialization tokens, making the process vacuous. However, when the vectors are initialized to random numbers, the nearest neighbors of the optimized vectors are the same for multiple relations.[5] As expected, unlike the corresponding soft prompts, these discretized prompts perform very poorly. For example, when training on BERT$_{BASE}$, OptiPrompt soft prompts achieve 48.26% T-ReX accuracy. The corresponding discretized prompts only achieve 1.21% accuracy.

---

[5]This might be related to the "prompt waywardness" phenomenon observed by Khashabi et al. (2021), who showed that, for any arbitrary embedding, it is possible to find a well-performing soft prompt that has that embedding as its nearest neigbhor.

Table 6: Comparison of LPAQA and AutoPrompt across LMs. Each method is trained on the LM leading to the best same-source-and-target performance.

| Source / Target | LPAQA BERT$_{\text{LARGE}}$ | AutoPrompt BERT$_{\text{BASE}}$ |
|---|---|---|
| BERT$_{\text{BASE}}$ | 38.01 | **50.09** |
| BERT$_{\text{LARGE}}$ | 42.14 | **47.01** |
| DistilBERT | 7.24 | **15.75** |
| RoBERTa$_{\text{BASE}}$ | 27.68 | **32.31** |
| RoBERTa$_{\text{LARGE}}$ | 33.68 | **37.79** |
| DistilRoBERTa | 22.21 | **31.71** |
| GPT2 | **7.88** | 4.70 |
| GPT2$_{\text{MEDIUM}}$ | **14.88** | 14.38 |
| GPT2$_{\text{LARGE}}$ | 17.15 | **17.95** |
| GPT2$_{\text{XL}}$ | 18.01 | **18.34** |
| BART$_{\text{BASE}}$ | 26.74 | **28.98** |
| BART$_{\text{LARGE}}$ | **28.47** | 26.73 |
| T5$_{\text{SMALL}}$ | **16.38** | 15.78 |
| T5$_{\text{BASE}}$ | 27.76 | **29.26** |
| T5$_{\text{LARGE}}$ | 31.20 | **32.32** |
| Average | 23.96 | **26.87** |

## C  LPAQA GENERALIZATION

AutoPrompt is better than LPAQA in the same-source-and-target setup, as demonstrated in Table 2. Table 6 further shows that AutoPrompt also generalizes better than LPAQA to other target models (we train the latter on BERT$_{\text{LARGE}}$, since this is the LPAQA setup reaching the best same-source-and-target accuracy).

## D  LAMA-UHN RESULTS

LAMA-UHN (UnHelpfulNames) (Poerner et al., 2019) is a subset of LAMA where facts that can be answered based on entity names alone were removed. The dataset is built using two heuristics. The first one filters facts whose object is a case-insensitive substring of the subject entity name. The second heuristic deletes a triple if a given model can infer information such as native language, nationality or place of birth using the subject's surface form only. We use the original dataset with BERT$_{\text{BASE}}$ as filtering model.

Table 7 shows that the generalization results still hold. The difference between BERT$_{\text{BASE}}$ and BERT$_{\text{BASE}}$/T5$_{\text{LARGE}}$ is smaller, possibly due to a ceiling effect with this harder data-set, but (once we exclude the non-generalizing same-source-and-target BERT$_{\text{BASE}}$ case), it is statistically significant (paired $t$-test, $\alpha$=0.05).

## E  CHOOSING MODELS FOR MIXED-TRAINING

We study to what extent same-target-same-source accuracies of component models are good predictors of mixture generalization accuracy. To avoid a combinatorial explosion, we fix BERT$_{\text{BASE}}$ (the best model in the same-target-same-source setup) as either generator or evaluator, and pair it with 15 different LMs. The correlation across 15 mixtures is at 0.22 when BERT$_{\text{BASE}}$ is fixed as generator, and at 0.49 when it is fixed as evaluator. Overall, this suggests that single-model setup performance is a good predictor of mixture quality. We attribute the lower score with BERT-as-generator to a mild ceiling effect in this case.

Indeed, we observe that, on average, performance is better when using BERT$_{\text{BASE}}$ as generator than as evaluator. This is likely due to the fact that the generator model does most of the "heavy lifting" in the mixture, as it is the one finding candidate prompts, that are then simply reranked by the evaluator. It thus makes sense to use the best single-model-setup system used as the generator in a mixture.

Table 7: AutoPrompt generalization evaluated on LAMA-UHN.

| Source / Target | $\text{BERT}_{\text{BASE}}$ | $\text{GPT2}_{\text{MEDIUM}}$ | $\text{T5}_{\text{LARGE}}$ | $\text{BERT}_{\text{BASE}}/\text{T5}_{\text{LARGE}}$ | $\text{BERT}_{\text{BASE}}/\text{GPT2}_{\text{MEDIUM}}$ | $\text{T5}_{\text{LARGE}}/\text{GPT2}_{\text{MEDIUM}}$ |
|---|---|---|---|---|---|---|
| $\text{BERT}_{\text{BASE}}$ | **39.80** | 13.69 | 15.00 | 31.03 | 28.61 | 13.98 |
| $\text{BERT}_{\text{LARGE}}$ | **37.91** | 17.88 | 20.73 | 32.00 | 32.25 | 13.81 |
| DistilBERT | **13.51** | 3.02 | 3.54 | 13.11 | 11.64 | 2.88 |
| $\text{RoBERTa}_{\text{BASE}}$ | 25.35 | 16.57 | 14.82 | **28.23** | 23.52 | 13.99 |
| $\text{RoBERTa}_{\text{LARGE}}$ | 30.82 | 18.91 | 20.14 | **31.62** | 28.18 | 18.14 |
| DistilRoBERTa | **24.98** | 14.17 | 11.71 | 24.61 | 21.03 | 13.01 |
| GPT2 | 2.54 | **6.65** | 3.68 | 6.48 | 3.83 | 5.22 |
| $\text{GPT2}_{\text{MEDIUM}}$ | 9.28 | 14.19 | 9.43 | 10.09 | **15.02** | 11.96 |
| $\text{GPT2}_{\text{LARGE}}$ | 11.63 | 10.88 | 9.78 | **15.06** | 13.30 | 10.87 |
| $\text{GPT2}_{\text{XL}}$ | 11.02 | 9.78 | 9.57 | 12.54 | **16.36** | 13.48 |
| $\text{BART}_{\text{BASE}}$ | 22.25 | 18.81 | 15.18 | **26.21** | 23.19 | 14.63 |
| $\text{BART}_{\text{LARGE}}$ | 19.62 | 18.74 | 13.80 | **25.07** | 21.24 | 14.53 |
| $\text{T5}_{\text{SMALL}}$ | 6.04 | 7.79 | 3.83 | **9.55** | 7.86 | 3.30 |
| $\text{T5}_{\text{BASE}}$ | 20.36 | 13.71 | 15.19 | **25.98** | 21.19 | 11.74 |
| $\text{T5}_{\text{LARGE}}$ | 23.46 | 20.70 | 31.48 | **34.25** | 25.84 | 19.37 |
| Average | 19.90 | 13.70 | 13.19 | **21.72** | 19.54 | 12.06 |

Finally, when fixing $\text{BERT}_{\text{BASE}}$ as generator, we observe that, in the majority of cases, we obtain better generalization results by using a different evaluator than $\text{BERT}_{\text{BASE}}$ itself, suggesting that mixed-training generalizes better compared to single model training even when randomly selecting one of the two models.

# F PROMPT EXAMPLES

Table 8 shows manual (LAMA) and AutoPrompt prompts for a randomly picked set of T-ReX relations. We present examples for AutoPrompt trained on $\text{BERT}_{\text{BASE}}$, the best choice in the same-source-and-target setup, and for mixed training using $\text{BERT}_{\text{BASE}}$ as generator model and $\text{T5}_{\text{LARGE}}$ as evaluator ($\text{BERT}_{\text{BASE}}/\text{T5}_{\text{LARGE}}$), since this combination results in the best generalization performance. For comparison, we also present prompts obtained using $\text{GPT2}_{\text{MEDIUM}}$ as source model, since this is a setup with relatively low performance in both the same-source-and-target and generalization evaluations.

We note that the two better models generally produce prompts that have at least some degree of semantic/lexical relevance to the relation. This might be to some degree the case for $\text{GPT2}_{\text{MEDIUM}}$ as well, but much more opaquely so. Interestingly, for the two better models at least, the most transparent information is concentrated towards the end of the string. For example, the $\text{BERT}_{\text{BASE}}$ *location* prompt ends with *headquartered in*, and the corresponding $\text{BERT}_{\text{BASE}}/\text{T5}_{\text{LARGE}}$ prompt ends with *headquarters in*. The $\text{BERT}_{\text{BASE}}$ *manufacturer* prompt ends with *marketed by*, and the corresponding $\text{BERT}_{\text{BASE}}/\text{T5}_{\text{LARGE}}$ prompt ends with *maker*. Both these observations (higher semantic relevance of better-performing prompts and higher information concentration at the end of the prompt string) are confirmed by the quantitative analyses reported in Section 4.4 of the main text.

By qualitative inspection, the only noticeable difference between the $\text{BERT}_{\text{BASE}}$ prompts (better in the non-generalizing setup) and the $\text{BERT}_{\text{BASE}}/\text{T5}_{\text{LARGE}}$ ones (better at generalization) is that the latter contain a higher proportion of well-formed English words (as opposed to concatenations of word pieces that do not result in existing words). Again, this is quantitatively confirmed by the analysis in Section 4.4

| relation | Manual | BERT$_{\text{BASE}}$ | GPT2$_{\text{MEDIUM}}$ | BERT$_{\text{BASE}}$/T5$_{\text{LARGE}}$ |
|---|---|---|---|---|
| *place of death* | [X] died in [Y] | [X]lland flees exilelessly downtown [Y] | [X] pseudonym dual (Paris and [Y] | [X] teaches modelling frescoes downtown in [Y] |
| *religion* | [X] is affiliated with the [Y] religion | [X] believe unidentified fundamental religion opposite [Y] | [X]":"/gnu\u9f8d\ufffd constitutes orthodox [Y] | [X] dynasty mosques helped peacefully spread [Y] |
| *location* | [X] is located in [Y] | [X] contrasts slalomgraphers headquartered in [Y] | [X] infographic biking & accommodation in [Y] | [X] specialisedingly based headquarters in [Y] |
| *country of origin* | [X] was created in [Y] | [X] comes versa publishedtoon outside [Y] | [X]estyle rivalry[Germany and [Y] | [X] streamed world semifinal badminton representing [Y] |
| *manufacturer* | [X] is produced by [Y] | [X] isn altitudes binary marketed by [Y] | [X] 650wagenJohnsonasakiphabet [Y]. | [X] devised by billion fielding maker [Y] |

Table 8: Example prompts for a random T-ReX relation subset. The Manual column shows manually-crafted LAMA prompts. The following two columns show prompts generated by AutoPrompt using as source models BERT$_{\text{BASE}}$ and GPT2$_{\text{MEDIUM}}$, respectively. The last column shows prompt generated through AutoPrompt mixed training, using BERT$_{\text{BASE}}$ as generator model and T5$_{\text{LARGE}}$ as evaluator model. In all cases, [X] and [Y] stand for the subject and object slots.

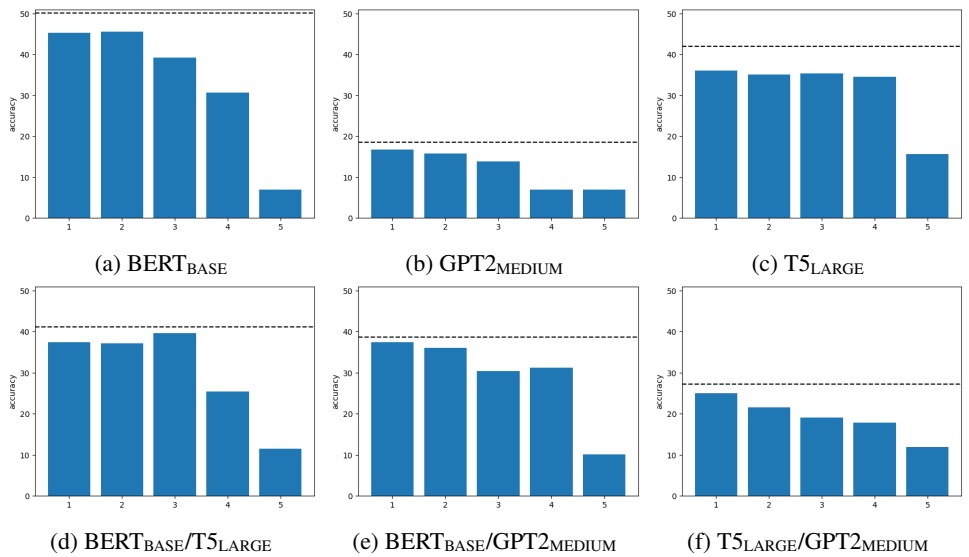

Figure 5: Percentage accuracy after dropping the token in each position of an AutoPrompt-generated 5-token prompt. The dashed horizontal line marks full-sequence accuracy.

## G TOKEN DELETION FULL RESULTS

Figure 5 reports the T-ReX accuracy for AutoPrompt prompt sets obtained using different LMs or LM combinations, always tested on the LM used for AutoPrompt training (the generator LM in mixed-training setups), when one token at a time is removed from the full prompt sequence.

