# OpenReview forum: "Can discrete information extraction prompts generalize across language models?"
_ICLR.cc/2023/Conference — ICLR 2023 poster_

### Official Review · Reviewer_9ydY · 2022-10-18

**Confidence:** 3
**Correctness:** 4
**Technical Novelty And Significance:** 3
**Empirical Novelty And Significance:** 4
**Recommendation:** 8

**Clarity, Quality, Novelty And Reproducibility:**

Clarity is excellent.

Minor Observation:
Table 2:  - please, indicate the used metrics in Table caption or title


Quality
* very rigorously performed study: from initial analysis of various approaches to prompting, to comparison of single LM-sourced eotoprompts to the introduction, comparison, and analysis of multi-LM-source autoprompts


Novelty
* the paper is original and novel
* the novelty is moderate with solid experimental grounding


Reproducibility
* to the best I could verify, it is reproducible

**Strength And Weaknesses:**

Strength:
* introduced a novel method and clearly showed its benefits and limitations
* thorough analysis of the potential underlying linguistic reasons (rather than treating as a black-box)
* clear narration
* comprehensive overview of related works
* solid grounding for experimentation

Weaknesses:
* while the paper doesn't present break-through findings, it is a solid and sufficiently impactful research

**Summary Of The Paper:**

The paper proposes a method  for automated prompt generation using a combination of Language Models. It shows that certain source LM combinations are more succesful than others in outperforming autoprompts trained on single ML.  The paper also includes a thorough analysis of potential reasons.

Additionally,  the paper provides an overview of the currentpormting approaches and povides their comparison, describing the recommended areas of use and limitations.

Contributions:
* a novel method for automated prompt generation from a mix of LMs
* comparison of promtpting approaches
* analysis of effects and limitations of the newly proposed method

I think that in addition to the proposed method, the paper may serve as great practical reference on prompting methods so it has  a potential for citation and community impact in various aspects.

**Summary Of The Review:**

It's an interesting solid paper with moderate novelty. Lean to accept

---

> ### Author Response · Authors · 2022-11-09
> **Response to Reviewer 9ydY**
>
> Thanks for your insightful and encouraging review. We will specify the Table 2 metric (micro-averaged percentage accuracy) in the table caption.

---

### Official Review · Reviewer_FEH5 · 2022-10-21

**Confidence:** 4
**Correctness:** 3
**Technical Novelty And Significance:** 1
**Empirical Novelty And Significance:** 3
**Recommendation:** 8

**Clarity, Quality, Novelty And Reproducibility:**

This work is clearly written and the message is evident to the reader without any struggle. The experiments are sound and the quality of the work is well done. The novelty of this paper does not lie in developing new models, but rather investigating and understanding the behaviour of existing models. This is a valid and useful idea and it contributes to the field of research.

**Strength And Weaknesses:**

- The motivation of this paper is clear and a worthwhile topic to investigate. The experiment setup and settings are good and the evaluation is sound. The authors provide ample analysis on their findings.

- The way this paper trains and evaluates, it would need to learn a different prompt for each relation in the slot-filling task. The authors use the original (manual) prompts provided by LAMA in their experiments. Isn't that limiting the findings? Have the authors looked into other prompt templates?

- The authors look at three types of language models and various prompt induction methods. The model they choose however is on the [relatively] smaller side. Do the authors think model size and the number of parameters is also a contributor to how these models learn and output knowledge?  Have the authors looked into other bigger models and if not what is their hypothesis about these model's behaviour and how their findings generalize to larger models?

- Table 3 shows that by simply mixing the models we would not achieve good generalization. Do the authors have a hypothesis why that's the case or have they looked further into it?

**Summary Of The Paper:**

This paper studies whether prompts that effectively extract information from a language model can also be used to probe other language models for the same information. The authors show that this is indeed not the case. They propose an approach to induce prompts by mixing language models at training time and show that in this way they can generalize well across models. They evaluate on the slot-filling task which is good to investigate the knowledge contained in language models. They provide extensive analysis and insight into the generalization of these large language models and how to effectively probe information from them.


**Summary Of The Review:**

This work is well-motivated, clearly explained, and sufficiently supported by analysis. There are some questions for the authors (see section Strength And Weaknesses), however overall it's a valuable contribution to the field.

---

> ### Author Response · Authors · 2022-11-09
> **Response to Reviewer FEH5**
>
> Thanks for your very helpful feedback and the suggestions. We will incorporate all the insights and new results reported here in the paper revision.
>
> ### T-ReX/LAMA task and prompts
>
> Concerning your points about the T-ReX/LAMA setup, we chose this task since it involves a large number of semantically contentful relations (41), as opposed to other tasks used in the prompting literature, where a single prompt (presumably simply describing the task in a LM-understandable way) must be learned for the whole data-set (e.g., sentiment analysis, NLI, etc.). We agree however that the way ahead is to devise systems that can jointly learn adaptive prompts for multiple relations/tasks/inputs, dispensing with the need to run the induction algorithm for each relation separately. This is the main priority in our current work. We also would like to expand our analysis to other prompting systems and data-sets in future research.
>
> Coming to your specific question about looking at other prompt templates, we implicitly did that by also testing the LPAQA system. LPAQA generates and evaluates a large variety of human-interpretable prompts through paraphrasing and Web mining. When using LPAQA, we are de facto exploring 30 prompt formulations for each relation. These include many intuitively reasonable variations. For example, the tested LPAQA templates for the developer relation include (among others): [X] is created by [Y], [X] is designed by [Y], [X] was designed by [Y], [Y] has been developed by [X], [Y] is developed from [X], [Y] is the development of [X], [Y] developed through [X], [Y] is the evolution of [X]. We are thus confident that our results are robust to extensive template variation.
>
> ### Model size and generalization
>
> Concerning the issue of how model size affects learning: thanks for suggesting to look into this. We were for the time being not able to replicate our experiments with larger models, due to computational resource limitations. We are still trying, and we will update you here if we do succeed. However, we did look at the correlation between model size and ability to generalize for the models we analyzed in the paper, and the results are clear enough that we do not expect larger models would change the picture. In short, there is a clear negative correlation between model size generalization ability, as we detail below.
>
> To take absolute model performance into account when looking at generalization, we computed a generalization-drop score (but similar results are also obtained when looking at non-normalized generalization accuracies). The generalization drop score was calculated by averaging, for each source model, the corresponding Fig. 3 column values. That is, the generalization drop score measures the average magnitude of the drop in accuracy when prompts from a source model are tested on a different target, with respect to the original same-source-and-target accuracy. We found that there is a large correlation (.6) between performance drop and model size (in number of trainable weights). This effect might be partially due to covariance with model class: the left-to-right class performs poorly in general, and it includes the largest models; conversely, the masked-model class performs best, and it includes the smallest models. However, the model-class confound does not fully account for the correlation, as we systematically observe the same effect when controlling for model type: for example, the smaller RoBERTa-base has a smaller generalization drop than RoBERTa-large; T5-small has a smaller drop than T5-base; GPT-medium has a smaller drop than either GPT-large or GPT-XL; BERT-base has a smaller drop than BERT-large. We do not have a clear explanation for this correlation, but we think it is an intriguing observation that should be further studied in the prompt analysis literature.
>
> ### Picking models to mix
>
> If we understand your last point correctly, you are surprised that, as Table 3 shows, we have to be “picky” about which models to mix in order to get good generalization. We believe that the problem stems mainly from the fact that, due to the criteria we used to pick the models to mix, in 2 of the 3 combinations we explored, the evaluation model was GPT2-medium, a model that is not particularly good at our task to start with. In preliminary experiments we have just run, we saw that, indeed, when combining BERT-base with other good models, such as roBERTa-large and BART-base, we get good generalization performance, above that of the component models. So, a simple take-home would be that, if you want good mixed-training generalization, you should make sure that you mix models that perform well in the same-target-same-source setup.  We are currently running more systematic experiments with further combinations, and we will update you here when we get the results.

---

> > ### Author Response · Authors · 2022-11-16
> > **Further experiments on model mixtures**
> >
> > We have now run a more thorough analysis to verify the hypothesis that the same-target-same-source accuracies of the component models are significant predictors of mixture generalization performance. To avoid a combinatorial explosion, we built mixtures by fixing BERT-base (our best model in the same-target-same-source setup) as either generator or evaluator model, and testing all possible combinations with 15 different LMs.
> >
> > First, we observe that results are generally better when using BERT-base as generator than as evaluator. This is likely due to the fact that the generator model does the “heavy lifting” in the mixture, as it is the one used to actually find candidate prompts, that are then simply reranked by the evaluator. So, it pays off to use a very good model as generator.
> >
> > Coming more specifically to our question of to what extent same-target-same-source accuracy is a good predictor of (averaged) mixture generalization accuracy, we find indeed significant correlations between these two variables. The correlation across 15 mixtures is at 0.22 when BERT-base is fixed as generator, and at 0.49 when BERT-base is fixed as evaluator. We attribute the lower score with BERT-as-generator to a mild ceiling effect in this case.
> >
> > Finally, when fixing BERT-base as generator, we observe than, in the majority of cases, we obtain better generalization results by using a different evaluator than BERT-base itself, suggesting that, while the mixture strategy is more brittle than we would like it to be, it’s still likely to improve over using a single model, even when randomly selecting one of the two models.

---

> > > ### Comment · Reviewer_FEH5 · 2022-11-17
> > > **Authors response answered my questions.**
> > >
> > > Thanks for clarifying some points and running additional experiments to support your hypothesis. I believe including this information in the paper is a good addition.

---

> > > > ### Author Response · Authors · 2022-11-17
> > > > **thanks**
> > > >
> > > > Thanks! We are definitely planning to include our new experiments in the paper revision!

---

### Official Review · Reviewer_wUgX · 2022-10-25

**Confidence:** 3
**Correctness:** 4
**Technical Novelty And Significance:** 3
**Empirical Novelty And Significance:** 3
**Recommendation:** 6

**Clarity, Quality, Novelty And Reproducibility:**

The clarity, quality of the paper is good. The novelty is moderate and the paper is easy to reproduce with the extensive provided resources and detailed descriptions.

**Strength And Weaknesses:**

Pros: The authors present a systematic study of the extent to which language models (LM) query protocols, that, following current
usage, the authors call prompting methods, generalize across LMs.  The authors conduct the extensive analysis of automatically induced discrete prompts, tentatively identifying a set of properties characterizing the more general prompts, such as a higher incidence of existing English words and robustness to token shuffling and deletion. The paper is well-written and easy to understand. There are comprehensive experimental results presented by the authors.

Cons: The contribution of this paper to the new methodology for natural language processing is limited. As a result, the analysis in the paper is also limited by mainly utilizing the existing methods.

**Summary Of The Paper:**

The authors study whether automatically-induced prompts that effectively extract information from a language model can also be used, out-of-the-box, to probe other language models for the same information. After confirming that discrete prompts induced with the AutoPrompt algorithm outperform manual and semi-manual prompts on the slot-filling task, the authors demonstrate a drop in performance for AutoPrompt prompts learned on one model and tested on another. The authors introduce a way to induce prompts by mixing language models at training time the results in prompts that generalize well across models. The authors conduct an extensive analysis of the induced prompts, finding that the more general prompts include a larger proportion of existing English words and have a less order-dependent and more uniform distribution of information across their component tokens.

**Summary Of The Review:**

Please refer to the points in the above sections to improve the paper.

---

> ### Author Response · Authors · 2022-11-09
> **Response to Reviewer wUgX**
>
> Thanks for your review. We agree that our paper relies on an existing method, however we think it is useful and novel in the following respects (we will highlight these novelties in the paper revision):
>
> - It introduces and illustrates the discrete prompt model generalization problem;
> - It introduces a novel method (language-model mixing at training time) to mitigate the problem, that is easy to implement and, at least in principle, prompting-method agnostic;
> - It presents an extensive analysis of the prompts introducing analytical techniques that, again, can be used in the future independently of (discrete) prompt induction method.
>
> Prompt induction is currently a very active area of research, and while it is important to propose new algorithms in order to induce higher-quality prompts, we think it is equally important to conduct analysis-centered studies, such as ours, that consider new scenarios, identify new failure modes of existing methods, try to mitigate them, and attempt to better understand the nature of the generated prompts.
>
> Other reviewers have highlighted the novelty and usefulness of the paper as follows. Reviewer FEH5: “The novelty of this paper does not lie in developing new models, but rather investigating and understanding the behaviour of existing models. This is a valid and useful idea and it contributes to the field of research.” Reviewer 9ydY: “I think that in addition to the proposed method, the paper may serve as great practical reference on prompting methods so it has a potential for citation and community impact in various aspects.”

---

### Official Review · Reviewer_DprU · 2022-10-27

**Confidence:** 4
**Correctness:** 3
**Technical Novelty And Significance:** 2
**Empirical Novelty And Significance:** 2
**Recommendation:** 5

**Clarity, Quality, Novelty And Reproducibility:**

The paper is well organized and the content is clearly delivered. However, the paper does not has its own model or theoretical idea. It's more like an technical report instead of a research paper.

**Strength And Weaknesses:**

Strength: The paper is easy to follow.
Weaknesses: the work lacks novelty, and one can barely find origin idea proposed in the paper. Although the detailed analysis seems to make some sense, the content seems not enough for a regular research paper, even for application track.

**Summary Of The Paper:**

The paper verifies whether automatically induced prompts can use the same information to apply to other models. It turns out that automatically induced prompts by AutoPrompt [Shin et al. 2020] outperforms manual and semi manual methods in slot-filling tasks, and verifies that automatically induced prompts can learn from one model and test in another, and the effect is not good. In addition, also found that more general prompts often contain some attributes.

**Summary Of The Review:**

The paper verifies whether automatically induced prompts can use the same information to apply to other models. It turns out that automatically induced prompts by AutoPrompt [Shin et al. 2020] outperforms manual and semi manual methods in slot-filling tasks, and verifies that automatically induced prompts can learn from one model and test in another, and the effect is not good. In addition, also found that more general prompts often contain some attributes. The content of the paper is clearly stated, however, generally, the work lacks novelty. Although the detailed analysis seems to make some sense, the content is not enough for a ICRL research paper, even for application track.

---

> ### Author Response · Authors · 2022-11-09
> **Response to Reviewer DprU**
>
> Thanks for your feedback. We would like to comment on two aspects of your review. First, from reading it, we get the impression that you think analysis work should be confined to technical reports rather than proper research papers: we respectfully disagree with this view, as we argue below. Second, our paper does propose a set of original ideas, that we highlight below (and we will state more explicitly in the paper revision).
>
> ### On analysis-focused papers
>
> We believe that analysis work is very important, especially in a high-impact and poorly understood area such as language-model prompting. Making progress in understanding when and how current prompting methods work or fail seems to us as important as developing yet another prompting algorithm. Without analysis work, we cannot assess the robustness, safety and reliability of prompting methods. Analytical insights can also provide insights useful to new model development. We are clearly not alone in the community in holding this view, as a prompt analysis paper received an Outstanding Paper award at ACL 2022 (Lu et al, Fantastically Ordered Prompts and Where to Find Them). Indeed, other reviewers of our paper agree with this view. Reviewer FEH5 states: “The novelty of this paper does not lie in developing new models, but rather investigating and understanding the behaviour of existing models. This is a valid and useful idea and it contributes to the field of research.” Reviewer 9ydY adds: “I think that in addition to the proposed method, the paper may serve as great practical reference on prompting methods so it has a potential for citation and community impact in various aspects.”
>
> ### Novelty of our paper
>
> We noticed that in your review you do not mention one of the main novel contributions of the paper, namely a method to improve prompt generalization across models by model-mixing at training time. The method only requires modification of the training regime of an established algorithm (AutoPrompt), and it could, as such, also be applied to other prompting methods to improve their generalization performance. We see the simplicity and consequent potential generality of the method as a positive feature.
>
> Our main novel contributions are 1) to introduce and empirically illustrate the discrete prompt model generalization problem; 2) to introduce a method to mitigate the problem; 3) to present several new analytical methods to gain insights about automatically learned prompts (that, again, are method-agnostic).

---

### Decision · Program_Chairs · 2023-01-20

**Decision:**

Accept: poster

**Justification For Why Not Higher Score:**

The methods proposed are only slight modifications of prior work, so the work is a bit incremental.

**Justification For Why Not Lower Score:**

Generalization is important and this paper puts forward a solution that can generalize across LMs.

**Metareview: Summary, Strengths And Weaknesses:**


The paper puts forward a method for generating prompts automatically in way that prompts can generalize to new language models.
This is a worthy goal, and the proposed idea of mixing LMs is interesting. I do agree with some of the reviewers that the  novelty, and execution, analysis of this work is not as strong as a typical paper at ICLR.  Having said that, on a positive  note, the proposed approach of mixing language models at training time can spur future work along this lines.
For this reason, a recommendation of accepting the  paper as poster seems appropriate.


**Note From Pc:**

if the above contains the word "oral" or "spotlight" please see: "oral" presentation means -> notable-top-5% and "spotlight" means -> notable-top-25%. As stated in our emails, we are disassociating presentation type from AC recommendations

**Summary Of Ac-Reviewer Meeting:**

n/a